# Common Vole as a Focal Small Mammal Species in Orchards of the Northern Zone

**Vitalijus Stirkė, Linas Balčiauskas ***  **and Laima Balčiauskienė**

Laboratory of Mammalian Ecology, Nature Research Centre, Akademijos 2, LT 08412 Vilnius, Lithuania;
vitalijus.stirke@gamtc.lt (V.S.); laima.balciauskiene@gamtc.lt (L.B.)

* Correspondence: linas.balciauskas@gamtc.lt or linas.balciauskas@gmail.com; Tel.: +370-685-34141

**Abstract:** In 2018–2020, we performed a country-wide study of small mammals in commercial orchards and berry plantations with the aim of determining whether the common vole (*Microtus arvalis*) is a more suitable focal species than the field vole (*M. agrestis*) in the risk assessment of plant protection products in Lithuania (country of the Northern Zone). Common vole was present in 75% of orchards and in 80% of control habitats, accounting for 30% of all trapped individuals. The proportion of this species was stable between years and seasons. The pattern was in agreement with the intermediate disturbance hypothesis, i.e., highest in medium-aged crops, while lowest in habitats with high intensities of agricultural practices. The average relative abundance of common vole in autumn, $2.65 \pm 0.52$ individuals per 100 trap days, was three times higher than that in summer, with no differences recorded between crops and control habitats. Field vole was present in 30% of locations, only accounting for 2.1% of all trapped individuals. In central and eastern European countries, common vole is more widespread and abundant than field vole. In Lithuania, common vole dominates in orchards and natural habitats and is, therefore, the most relevant small mammal species for higher tier risk assessment.

**Keywords:** risk assessment; focal species; voles; orchards; berry plantations; Northern Zone; Lithuania


## 1. Introduction

Accounting for a quarter of the mammal species in Lithuania [1], and for over one-third of known mammal species in the world [2], rodents are the most species-rich group of mammals. Rodents are characterized by a variety of diets [3,4], these indicating the diversity of inhabited habitats, including agricultural lands [5–8]. As agricultural habitats, such as cropland, orchards, and small-scale farming, cover a significant part of Europe [9], they are a very important source of rodent diversity [10–12].

The presence of rodents in agricultural lands, however, pose a dilemma. Rodents provide ecosystem functions in the agricultural landscapes [13] and are a food source for many carnivorous mammals and birds of prey [14,15], thus are important for the functioning of a healthy food web [13,16,17]. As a result, declining diversity in European agricultural landscapes [18] have resulted in measures to enhance land capacity for wildlife [19], such as the Entry Level Scheme Tier of Environmental Stewardship [20]. However, the presence of rodents in agricultural landscapes also has negative consequences, including crop damage. Two *Microtus* species, the common vole (*Microtus arvalis*) in western Europe and the field vole (*Microtus agrestis*) in eastern Europe [21–23], along with the wood mouse (*Apodemus sylvaticus*) and the bank vole (*Clethrionomys glareolus*) in northern Europe [3,7], are listed as main crop pest species. Most of these species are also listed as forest pests [24].

Conventional pest management measures, such as rodenticides, have several drawbacks. First, they reduce the biological diversity of all small mammals, killing species that do no damages to crops (i.e., [25]). Secondly, rodenticides also poison carnivorous and other animals (including granivorous birds), concentrating in their bodies [26–28]. Likewise, negative consequences to animals can be caused by plant protection agents such

as pesticides. Based on the requirement that "information should be provided to enable an assessment of the direct impact on birds and mammals likely to be exposed to the active substance, plant protection product and/or its metabolites" [29], focal species were selected to represent the various agricultural habitats of European Union (EU) Member States. Originally, for the various crops, including orchards and meadows, the focal species for small mammals were the insectivorous common shrew (*Sorex araneus*), the herbivorous common vole, and the omnivorous wood mouse. Depending on the habitat, the last species was referred to as insectivorous, herbivorous, omnivorous or granivorous [29]. In the latest pesticide risk assessment, the focal species were selected according to geographic zones, whereby Lithuania is included within the Northern Zone [30]. While field vole was referred to as the representative herbivore and wood mouse as the omnivore species for small animals, a possible exception is outlined for the Baltic countries. Common vole is given as a possible substitute for the field vole, and striped field mouse (*Apodemus agrarius*) as a possible substitute for wood mouse [30].

For Lithuania, the selection of the small mammal focal species is relevant for several reasons. To begin with, wood mouse is quite rare and not related to agricultural habitats, while field vole is also not a best representative for this land type ([1,12,31–34], see Discussion for details). Furthermore, prior to the pilot study [12], small mammal communities in the agricultural lands of the Baltic countries had not been investigated. Given current changes within Lithuanian agriculture, specifically a 12.1% decrease in stock-raising and an 8.9% decrease in the area of meadows and pastures between 2015 and 2020, the result has been an increase in the area of cropland [35], thus the choice of the focal rodent species requires additional data. In the 2015–2019 period, the area of commercial orchards and berry plantations was fairly stable at about 30,000 ha. Sown area between 2015 and 2020 increased from 2,081,051 ha to 2,144,873 ha, while area of meadows and pastures decreased, from 798,926 ha to 728,018 ha, respectively [35]. Orchard habitats, though a significant source of small mammal diversity in the agrolandscape [12], had not previously been analyzed as a habitat for the focal species in the Baltic countries.

Therefore, in this study, we aimed to determine which species of small herbivores could be used as a focal species for risk assessment of plant protection products in Lithuania. Specifically, we tested whether common voles or field voles were widespread and abundant enough in fruit orchards, berry plantations, and adjacent control habitats. Based on long-term small mammal investigations in the country [1] and preliminary results of their trapping in the orchards [12], our working hypothesis stated that the distribution and abundance of field voles was not sufficient to classify this as a focal species.

## 2. Material and methods

### 2.1. Study Sites

In 2018–2020, we investigated small mammals at 20 trapping locations within 18 study sites across Lithuania (northern Europe) (Figure 1a). Covering a number of apple and plum orchards, as well as currant, raspberry, and highbush blueberry plantations, each site had a respective control habitat (mowed meadow, unmoved meadow, or forest ecotone) at a nearby vicinity. Sites 1–3, 6–10, and 12 were investigated in 2018–2020, sites 5, 7, 9, 11, 13–15 in 2018–2019, and sites 16–18 in 2020.

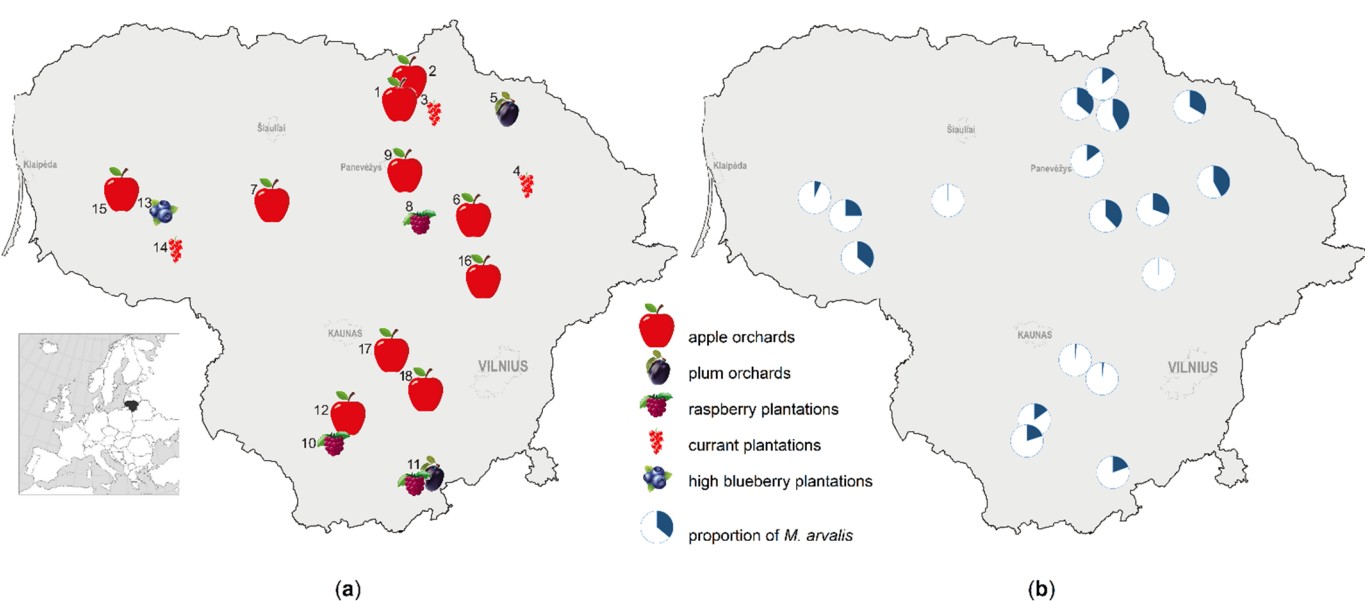

**Figure 1.** Location of the study sites in Lithuania with an indication of the crops (**a**) and the proportion of common voles in the small mammal communities in the orchards, plantations, and control habitats at these sites (**b**).

Investigation sites were characterized by different crop ages (young, medium-aged, or old) and different intensities of agricultural practices. Depending on the soil scarification, grass mowing, mulching of the plant interlines, and usage of the rodenticides and plant protection agents, we characterized three levels of intensity. Sites with only grass mowing once or several times per season were attributed to low intensity, while usage of two measures from those above listed once or twice per season were defined as medium intensity. Application of several measures or frequent application of two measures per season was defined as high intensity. The distribution of the study sites according crop age and intensity of agricultural measures is shown in Table 1.

**Table 1.** Distribution of the study sites and trapping effort (trap days) according to crop age, intensity of agricultural practices, and control habitats.

| Parameter | Values | Sites | Trapping Effort |
|---|---|---|---|
| Crop age | old | 1,2,6,7,9,12,16–18 | 9768 |
| | medium | 3,4,8,11,13–15 | 5050 |
| | young | 1,5,10,12 | 1900 |
| Intensity of agriculture | high | 2,6,10–13,15,17 | 8218 |
| | medium | 1,5,9,14 | 4450 |
| | low | 3,4,7,8,11,16,18 | 4050 |
| Control | forest | 11,17 | 525 |
| | mowed meadow | 1,2,4,6,8–10,12,13–16 | 5560 |
| | non-mowed meadow | 1,3,5,7–9,11,18 | 2700 |

## 2.2. Small Mammal Trapping

We snap-trapped small mammals using standard method [36], lines of 25 traps set at 5 m intervals, with 1 to 4 lines per habitat depending on the size of orchard, and 2 lines per control. Traps were exposed for three days, checked once per day. We used brown bread and raw sunflower oil for the bait, changing it after rain or when consumed. Total trapping effort was 25,503 trap days, divided between 16,718 for the orchards and 8785 for the control habitats (Table 1).

Snap traps may have different performance in trapping shrews compared to the pitfall traps [36], however we were not able to use pitfall traps in the orchards and berry fields due to agricultural activities.

Common voles and field voles were identified by their teeth at dissection and after cleaning skulls [37].

*2.3. Data Analysis*

The proportion of common vole and the 95% CI for species proportion among all trapped small mammals were calculated with the Wilson method of the score interval [38] using OpenEpi epidemiological software [39]. Differences in the proportions of common vole between habitats, crop ages, and intensities of agricultural measures were evaluated using G test was used using online calculator [40]. Effect size was expressed according to adjusted Cohen's w [41], calculated in WinPepi, version 11.39 (Abramson, J., Jerusalem, Izrael).

Diversity index, Shannon's H, was calculated in PAST version 4.01 (Paleontological Museum, University of Oslo, Oslo, Norway). Normality of distribution of relative abundances was tested using Kolmogorov–Smirnov test. We found mixture of patterns, i.e., not all data were distributed normally. We, however, rely on the one-way ANOVA being a robust test against the normality assumption, tolerating violations rather well. Therefore, to test the influence of the year, season, crop type, crop age, and intensity of agricultural practices, we applied main effects ANOVA to common vole relative abundance, expressed as individuals per 100 trap days, using Tukey HSD with unequal N for post-hoc analysis. The confidence level was set as $p < 0.05$. Calculations were done in Statistica for Windows, version 6.0 (StatSoft, Inc., Tulsa, OK, USA).

**3. Results**

In the period 2018–2020, we trapped 1449 small mammals in the orchards, plantations, and control habitats. Eleven species were identified: common shrew, pygmy shrew (*Sorex minutus*), house mouse *(Mus musculus)*, harvest mouse (*Micromys minutus*), yellow-necked mouse (*Apodemus flavicollis*), striped field mouse, and five vole species, including common vole, root vole (*Microtus oeconomus*), field vole, bank vole and water vole (*Arvicola amphibius*). Diversity of small mammals in the commercial orchards (10 species, H = 1.54) was lower than that in the control habitats (11 species, Shannon's H = 1.76, $p < 0.001$). Out of all individuals 436 were identified as common voles, a proportion of 30.1% (CI = 27.8–32.5%). This species was present in 75% of trapping locations in the orchards and plantations (absent in sites 7, 10 13, 15, and 17) and in 80% of control habitats (absent in sites 9, 11, 16, and 18). Common vole was the dominant or subdominant species (Table 2), along with the yellow-necked mouse and striped field mouse.

**Table 2.** Dominance pattern of common vole in the studied sites, 2018–2020, with species proportions shown in parentheses.

| Season | Species | 2018 | 2019 | 2020 |
|---|---|---|---|---|
| Summer | Dominant | *M. arvalis* (27.2%) | *M. arvalis* (38.3%) | *A. flavicollis* (39.7%) |
|  | Sub-dominant | *A. flavicollis* (20.7%) | *A. flavicollis* (27.1%) | *M. arvalis* (26.9%) |
| Autumn | Dominant | *A. agrarius* (37.1%) | *M. arvalis* (36.2%) | *A. agrarius* (32.4%) |
|  | Sub-dominant | *M. arvalis* (25.7%) | *A. flavicollis* (29.9%) | *A. flavicollis* (30.3%) |

Field vole, currently proposed as one of small mammal focal species in the country, was represented by only 31 individuals, a species proportion of 2.1% (95% CI = 1.5–3.0%). Field vole was present in 30% of trapping locations (20% of orchards, not registered in berry plantations). It was not numerous in any of the orchards—the maximum was eight individuals at site 9 during the 2018–2020 period. Field vole was present only in old apple orchards (with no regard to the intensity of agricultural practices) and, more so, in the mowed meadows.

The number of common shrews, being the focal species for insectivorous animals proposed for Lithuania, was even smaller—we trapped 27 individuals during the 2018–2020 period, the proportion being 1.9% (CI = 1.3–2.7%), though they were present in 60% of trapping locations and 25% of investigated orchards.

The third proposed focal species, wood mouse, was not trapped in the orchards, berry plantations, or their control habitats.

No small mammals were trapped in the high blueberry plantation.

### 3.1. Share of Common Vole in Small Mammal Communities

The proportion of common vole in the small mammal communities was rather stable. In 2018 their proportion was 26.0% (95% CI = 22.4–30.0%), in 2019 it was 36.7% (32.9–40.1%), and in 2020 it was 25.4% (21.2–30.1%) of all individuals. The increase in 2019 was not strongly appreciable (comparing to 2018, G = 13.99, $p < 0.05$, Cohen's w = 0.164, small effect size, comparing to 2020 G = 0.01, NS, w = 0.167, small effect size). The proportion of common vole in the summer seasons of 2018–2020 was 32.0% (CI = 27.0–37.5%), while in the autumn it was 26.0% (27.0–32.3), the difference not expressed (G = 0.56, NS, w = 0.031, no effect). The proportions of common vole by location, based on pooled data, are presented in Figure 1b.

We found common vole proportion to be higher in the orchards and plantations than in the control habitats, and the respective figures were 38.0% (CI = 34.8–41.4%) and 18.8% (15.9–22.2%) of all small mammals, with the difference being significant (G = 63.0, $p < 0.001$, w = 0.360, medium effect size). Proportions according to crops and control habitats are presented in Figure 2a,b.

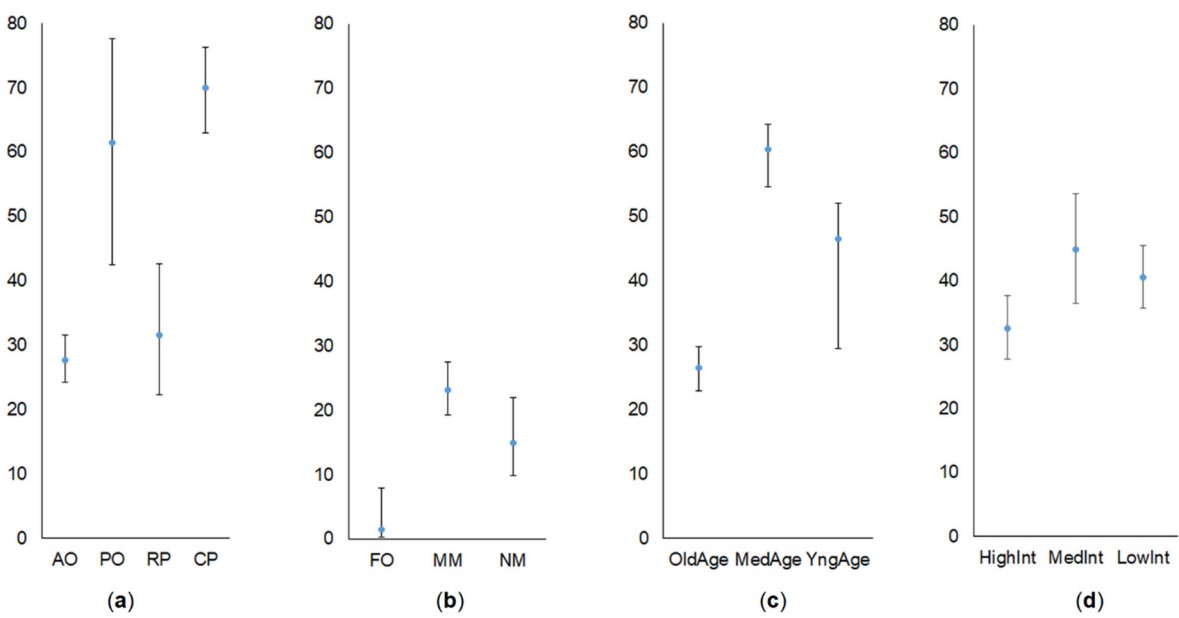

**Figure 2.** Proportions of common vole (in %) in commercial orchards and berry plantations (**a**), control habitats (**b**), according to the crop age (**c**) and intensity of agricultural practices (**d**). Whiskers denote 95% CI. Habitats: AO—apple orchards, PO—plum orchards, RP—raspberry plantations, CP—currant plantations, FO—forest, MM—mowed meadow, NM—non-mowed meadow.

The proportion of common vole in the currant plantations (Figure 2a) exceeded that in apple orchards (G = 108.5, $p < 0.001$, w = 0.569, large effect size) and raspberry plantations (G = 30.99, $p < 0.001$, w = 0.539, large effect size), being the same as in plum orchards (G = 0.41, NS, w = 0.086, no effect). The proportion of the species in apple orchards was smaller than in plum orchards (G = 10.9, $p < 0.001$, w = 0.218, small effect size), but did not differ from that in raspberry plantations (G = 0.32, NS, w = 0.040).

In the control habitats, common vole proportion was highest in mowed meadows (Figure 2b), exceeding that in non-mowed meadows (G = 3.76, $p < 0.05$, w = 0.124, small effect size). Significantly, the smallest proportion of the species was observed in forest controls (compared to moved meadows, G = 22.2, $p < 0.001$, w = 0.275, small effect size; compared to non-mowed meadows, G = 9.22, $p < 0.01$, w = 0.301, medium effect size).

The pattern of common vole proportions according to the crop age (Figure 2c) and according to intensity of agricultural practices (Figure 2d) were in agreement with the intermediate disturbance hypothesis. Species proportion was highest in the middle age crops, significantly exceeding that in the old (G = 87.2, $p < 0.001$, w = 0.495, medium effect size) and insignificantly in young (G = 1.50, NS, w = 0.117, small effect size) crops. The common vole proportion in young crops was higher than in old ones (G = 4.0, $p < 0.05$, w = 0.137, small effect size).

Common vole avoided habitats with high intensities of agricultural practices (Figure 2d). Here, the species proportion was lower than in habitats with medium (G = 5.47, $p < 0.05$, w = 0.164, small effect size) and low (G = 4.59, $p < 0.05$, w = 0.117, small effect size) intensities of agricultural practices.

### 3.2. Abundance of Common Vole

The average relative abundance of common vole irrespective to the habitat in 2018–2020 was 1.72 $\pm$ 0.28 (95% CI = 1.17–2.28) individuals per 100 trap days. Time factor (year and season) explained an insignificant part of the abundance variation ($R^2 = 0.067$, $F_{3,164} = 4.97$, $p < 0.01$), with season being the only significant factor ($F_{1,164} = 11.08$, $p < 0.002$). The average abundance of common vole in autumn was threefold higher than that in summer, (2.65 $\pm$ 0.52 vs. 0.84 $\pm$ 0.19 ind. per 100 trap days, Tukey HSD, $p = 0.001$).

There were no significant differences in common vole abundance in orchards, plantations, or control habitats ($R^2 = 0.035$, $F_{7,160} = 1.87$, $p = 0.08$). The highest relative abundance was found in the currant plantations (Figure 3a) and mowed meadows (Figure 3b).

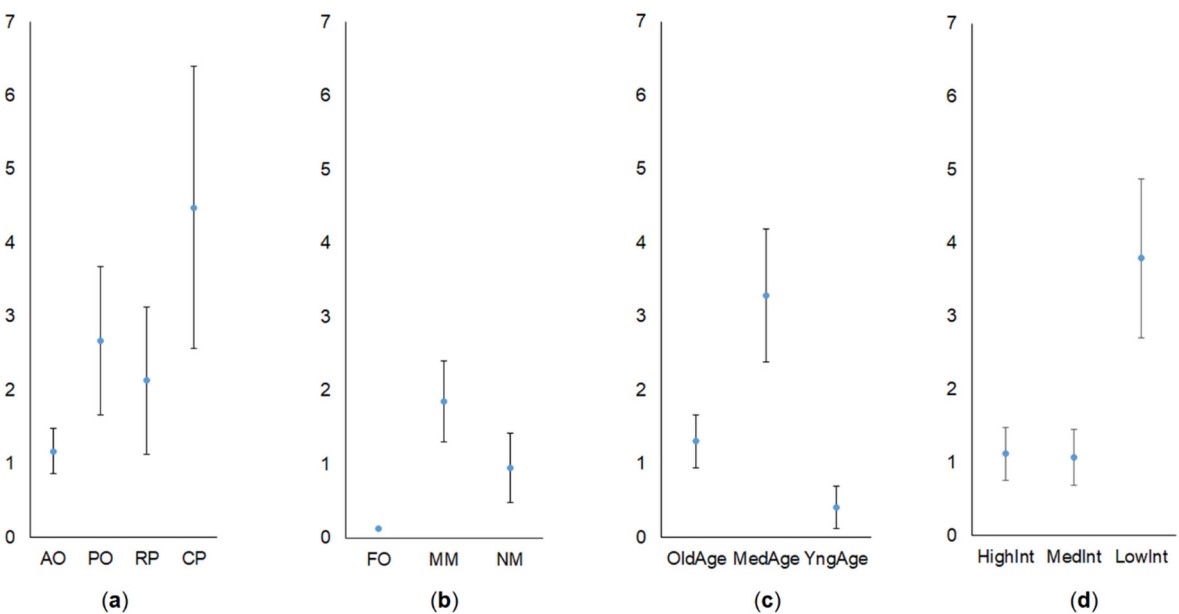

**Figure 3.** Relative abundance of common vole (individuals per 100 trap days) in commercial orchards and berry plantations (**a**), control habitats (**b**), according to the crop age (**c**) and intensity of agricultural practices (**d**). Whiskers denote SE. Habitats: AO—apple orchards, PO—plum orchards, RP—raspberry plantations, CP—currant plantations, FO—forest, MM—mowed meadow, NM—non-mowed meadow.

The cumulative influence of crop age and intensity of the agricultural practices on the relative abundance of common vole was significant, but not strong ($R^2 = 0.090$, $F_{4,81} = 3.10$, $p = 0.02$). The effect of the crop age ($F_{2,83} = 3.85$, $p = 0.025$) was less expressed than the effect

of the agricultural practices ($F_{3,164}$ = 3.85, *p* = 0.01). The relative abundance of common vole in the medium aged crops was ca eight times higher than in young crops and 2.5 times higher than in old crops (Figure 3c). The relative abundance of common vole in the crops with low intensity of agricultural practices was over threefold higher than that in medium or high intensity (Figure 3d; Tukey HSD, *p* < 0.025) and 2.5 times higher than that in control habitats (Figure 3b).

## 4. Discussion

Our results show that common vole was the dominant or subdominant rodent species in fruit orchards and berry plantations, with the proportion of species and abundance exceeding those in the adjacent control habitats, whereas field vole proportions were quite scarce. Data of the pilot study into small mammal diversity in commercial orchards [12] did not cover the issue of domination of different vole species. Furthermore, as the dominant species were not stable between 2018 and 2020 (see Table 2), we need a longer period of study to cover possible fluctuations. How do these data fit to the general picture of distribution of both species in Lithuania?

A summary of small mammal data from 58 sampling sites located in all parts of Lithuania, investigated in 1983 and 1991–1997, and representing various non-agricultural habitats, such as forests, wetlands, shrubby areas, reedbeds, and meadows [31] is in accordance with the results analyzed in the current paper. Common vole was present in 26 sites and its proportion in general was 7.9% (CI = 7.1–8.7%) of all trapped small mammals. Field vole was present in 34 sites, the difference not significant, with an average proportion of 5.7% (5.1–6.4%). Thus, common vole was more prevalent (G = 15.3, *p* < 0.001). The maximum proportions of these vole species was 80% and 48%, respectively. Field vole did not dominate any habitat type: Their proportion was up to 9.4% in wet forests and swamps, and 6.3% in open areas and meadows [42].

These findings correspond to other available data on long-term small mammal investigations in various habitats of Lithuania. In the north-eastern part of the country, in pooled data for forest, wetland, and meadow habitats in 1981–1990, field vole accounted for 2.7% (CI = 2.1–3.5%) of the 2349 trapped individuals, while common vole for 7.2% (CI = 6.3–8.4%), the latter species being significantly better represented (G = 44.1, *p* < 0.001; recalculated from [33]). Monitoring of 17 locations in eastern and north-eastern Lithuania in 2000–2005 showed the common vole proportion being 1.8% (CI = 0.8–4.1%, from 0 to 12.1% in different locations) in forest fragments and 67.6% (CI = 56.1–77.3%, from 0 to 95.3%) of all trapped individuals in the matrix of surrounding fields; field voles were not found [34].

In the Nemunas River Delta in west Lithuania, trapping in flooded and non-flooded meadows, agricultural fields, and flooded forest in 2004–2011 [43] also showed a higher prevalence of common vole, their proportion being 2.1% (CI = 1.5–2.8%), while that of field vole 0.4% (CI = 0.2–0.8%) (G = 25.2, *p* < 0.001).

In a succession of abandoned meadows to forest, monitored in 2007–2013 in the northern part of Lithuania, field vole proved to be more resistant to this kind of habitat change [32] – the proportion of common vole in the small mammal community decreased from 19.3% (CI = 16.1–22.9%) in the meadow, to 14.3% (11.7–17.4%) in young forest, and finally to 3.2% (2.0–5.1%) in advanced forest. The proportions of field vole were 12.5% (9.9–15.6%), 12.7% (10.2–15.7%) and 5.0% (3.4–7.2%), respectively. This pattern is in agreement with Wegge and Rolstad [44], also showing a decrease of field vole in older forests in Norway.

Based on the fact that owl diet reflects the composition of small mammal communities (e.g., [45–48]), this also confirmed in Lithuania [49], we checked data from several European countries on the issue of common vole / field vole presence in various habitats.

An absolute dominance of common vole (81.8%, CI = 81.4–82.2%) in the diet of long-eared owl (*Asio otus*) was found in a13-year-long study (*n* = 32,884) in agricultural areas of Slovakia [50], while common vole accounted for 80.2% of long-eared owl diet in winter

and 59.7% in summer in a fragmented forest-farmland landscape of central Poland [51]. *M. arvalis* also accounted for 86% of the diet of European kestrel (*Falco tinnunculus*) in Switzerland [48]. Likewise, in Hungary, common vole dominated (13–94%, depending on the cycle phase of the voles) the diet of the barn owl (*Tyto alba*) in farmland landscape [15]. Surprisingly, no field voles were registered in these large-scale studies.

Several other studies indicated a significant dominance of common vole. In a fragmented landscape composed of meadows, crops, and in lesser amount, forests in Poland, common vole accounted for 26.9% of the diet of barn owl and 74.1% of the diet of long-eared owl, the respective numbers of field vole being just 0.08% and 0.5% [52]. In the crop-area dominated landscape of central Poland, the common vole proportion in the diet of long-eared owl was 52.9–68.0%, while that of field vole only 0–0.1% [53]. In cornfield dominated agricultural landscape of Slovenia, where forests account for less than 20% of the area, the proportion of common vole and field vole in the diet of long-eared owl was 55.4% and 11.5% respectively [46].

From the presented data, we can presume a scarcity of field vole in agricultural areas of central and eastern Europe, which is in agreement with Romanowski and Żmihorski [53]. An increase in the percentage of forest in the landscape is correlated to an increase in the proportion of field vole, for example 22.0% (CI = 18.1–26.5%) against 1.1% (0.8–2.7%) of common vole in the diet of the Tengmalm's owl (*Aegolius funereus*) in the Czech Republic [47].

The abundance of common vole in various agricultural environments of different countries is confirmed by many authors ([54–56], and references therein). The species is well established in winter rape fields, these used as a main food source and thus suited to overwintering, this then resulting in damage to the crops [57]. In alfalfa fields, growth in common vole populations may become exponential during outbreak phases [58], resulting in high levels of damage [23,54]. This species is widely used in risk assessment for plant protection products ([22,59] and references therein).

In Scandinavian countries, field vole becomes dominant over common vole, especially in forested land, whereby densities of the former species regulate the breeding densities and other responses of owl and bird of prey species [45]. In Norway, the field vole proportion was highest in clearcuts (12.0%, CI = 7.8–18.2%), significantly higher than the species proportion in middle aged forest plantations (2.3%, CI = 0.4–12.1%) and old forest (1.5%, CI = 0.3–8.1%), Cohen's w = 0.191 and 0.245, respectively, effect size small (recalculated from Wegge and Rolstad 2018). In open fields of southern Sweden, field vole on average formed 71.1% of small mammals eaten by long-eared owl (83.0–92.9% in January). However, it was nearly absent in grazed fields [60].

All provided materials therefore confirm that despite Lithuania being considered part of the Northern Zone in terms of selecting the relevant small mammal species for higher tier risk assessment [30], the small mammal community structure is actually more related to that of central or eastern Europe. Our results show that common vole strongly dominated commercial orchards and adjacent control habitats, exceeding field vole in numbers and distribution. As a result, common vole is better suited as a focal species for small mammals in Lithuania. Small mammal research in the crop fields and other agricultural habitats of the country would be prospective to check, if common vole domination is more widespread.

**Author Contributions:** Conceptualization and investigation, L.B. (Linas Balčiauskas), V.S., and L.B. (Laima Balčiauskienė); methodology and formal analysis L.B. (Linas Balčiauskas); data curation, V.S. and L.B. (Laima Balčiauskienė); resources, V.S.; supervision, project administration and funding acquisition, L.B. (Linas Balčiauskas). All authors participated in writing draft, read and approved the manuscript.

**Funding:** In 2018 and 2019, this research was funded by the MINISTRY OF AGRICULTURE OF THE REPUBLIC OF LITHUANIA, grant number MT-18-3.

**Institutional Review Board Statement:** The study was conducted in accordance with Lithuanian (the Republic of Lithuania Law on the Welfare and Protection of Animals No. XI-2271) and European legislation (Directive 2010/63/EU) on the protection of animals and approved by the Animal Welfare Committee of the NATURE RESEARCH CENTRE, protocol No GGY-7. It Snap trapping was justifiable as we studied reproduction parameters and collected tissues and internal organs for analysis of pathogens, stable isotopes and chemical elements (not covered in this publication).

**Informed Consent Statement:** Not applicable.

**Data Availability Statement:** After publication, research data will be available from the corresponding author upon request. The data is not publicly available due to its usage in the ongoing study.

**Acknowledgments:** We thank Jos Stratford for checking the language.

**Conflicts of Interest:** The authors declare no conflict of interest. The funders had no role in the design of the study, nor in the collection, analysis or interpretation of data, or the writing of the manuscript or in the decision to publish the results.

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
