# Peer review of "Common Vole as a Focal Small Mammal Species in Orchards of the Northern Zone"

_diversity, doi:10.3390/d13030134_

Round 1
Reviewer 1 Report
The article: “Common vole Microtus arvalis as a reference small mammal species in the Northern Zone” from Lithuania is nice information on small mammal species in orchards and berry plantations with emphasis to two vole species Microtus arvalis and M. agrestis.
As there is a low share of agricultural land in Lithuania this study may be representative for common agriculture practice in this country and its small mammal assemblage management. A great effort has been done in three years study, with the result of underlining important role of a common vole in this country agriculture. There is no information from this part of Europe which species are in agriculture landscape. So the information is valuable.
I have some suggestions and corrections to this study.
I think the table with all small mammal species trapped and their diversity index would enrich the data and show more of this country diversity in agriculture landscape. There may be also other species of mice as Apodemus uralensis, which is in Central Europe typical for open and agriculturally used landscape. Presence of the other vole species as Microtus mystacinus would be also very interesting (See: Wilson , D. E., Lacher, T. E., Jr. and Mittermeier, R. A. eds. (2017) Handbook of the mammals of the world. Vol. 7. Rodents II. Lynx Edicions, Barcelona.).
As to identification of the species e.g. M. agrestis – M. arvalis there is much easy way to identify species immediately after trapping by morphological differences between voles as their body measurements, ear and hind leg foot measurement and so, but your way may be precise.
As to figure 2 and 3 identification of graphs (a) to (d) would be better to put the letter up on left side of coordinates y.
Reference 10: Names are opposite: Alain is name Butet is surname and so on.
Author Response
Rev#1 comments and answers
Thank you for the positive comments, and please find our answers below, point to point
Comment
I think the table with all small mammal species trapped and their diversity index would enrich the data and show more of this country diversity in agriculture landscape. There may be also other species of mice as Apodemus uralensis, which is in Central Europe typical for open and agriculturally used landscape. Presence of the other vole species as Microtus mystacinus would be also very interesting (See: Wilson , D. E., Lacher, T. E., Jr. and Mittermeier, R. A. eds. (2017) Handbook of the mammals of the world. Vol. 7. Rodents II. Lynx Edicions, Barcelona.).
Answer
Presence of Microtus mystacinus is not possible in Lithuania; Apodemus uralensis was so far not trapped in the agricultural habitats. In our country, “the majority of specimens were captured in the ecotones of mixed forests and open habitats (natural meadows, cornfields and fallow fields) and in open habitats bordering forests or situated close to them”: Juškaitis, R., Balčiauskas, L., & Alejūnas, P. (2016). Distribution, habitais and abundance of the herb field mouse (Apodemus uralensis) in Lithuania. Biologia, 71(8), 960-965.
We intend to publish small mammal diversity data as separate paper in next year, therefore presenting table with all trapping data would be premature; addressing your comment, we add species list and diversity evaluation as text.
Comment
As to identification of the species e.g. M. agrestis – M. arvalis there is much easy way to identify species immediately after trapping by morphological differences between voles as their body measurements, ear and hind leg foot measurement and so, but your way may be precise.
Answer
We had not only these Microtus species, but also M. oeconomus, making field identification more difficult; hair on the edge of an ear, for example, did not work. Anyway, all specimens were dissected at the laboratory, therefore we used 100% reliable identification by their teeth.
Comment
As to figure 2 and 3 identification of graphs (a) to (d) would be better to put the letter up on left side of coordinates y.
Answer
We labelled figures 1–3 as required by the journal – labels in the bottom part and centered.
Comment
Reference 10: Names are opposite: Alain is name Butet is surname and so on.
Answer
Corrected; it still is mistakenly reported by the Google Scholar, and in the journal mane and surname were in wrong places. We do not know any of these authors. If it was you, our apologies.
Reviewer 2 Report
Dear Authors,
The topic of the manuscript is relevant to the goals of the risk assessment of plant protection products, because the study examines the reference role of potential pest rodents and is focused on the importance of common vole (Microtus arvalis) for the risk assessment of rodenticides and plant protection agents. The manuscript is concise and reads well, and the results are explained clearly.
The strength of the manuscript is that the reference problem of potential pest species is well-introduced, so the goals of the manuscript are clear. In addition, this work is country-wide and the control areas were chosen appropriately. However, in the introduction the agricultural land transformation is described (line 67-72), but the Authors did not give details (e.g. area) of the agricultural lands other than orchards and fruit plantations (e.g. cereal crops). The title of the paper refers to a wider range of habitats than those investigated in the study. In the discussion, many of the cited studies were conducted in crop-dominated areas and the conclusion in line 297-298 refers to the whole area of Lithuania. Therefore, I suggest including some information (e.g. area) on other agricultural fields (crops) in the introduction. Based on the title and the discussion, and in the light of the findings in this study, a short evaluation of similar studies in cereal fields of Lithuania would be relevant in the discussion. Alternatively, the Authors could give suggestions for future research in crop fields.
The methods of data procession, the applied statistical methods are appropriate. However, Chi-square test is generally used to compare count data, and is not recommended to compare proportions. To test differences in percentage proportions, I recommend using G-test, or proptest in R. In the effect test of categorical predictor variables (e.g. crop age) in main effects ANOVA, I suggest to include the results of normality tests.
In the material and methods, the Authors wrote they used Hotelling’s T-square for multivariate analyses (MANOVA). However, in the section 3.2 (beginning in line 188), MANOVA results are not included (e.g. Hotelling’s T-square probabilities are not included). The results indicate that main effects ANOVA was used, and not multivariate ANOVA (MANOVA). This is correct, because if only one dependent variable was examined (proportions of M. arvalis), main effects ANOVA is appropriate. Therefore, I suggest changing the text accordingly (remove the description of Hotelling’s T-square for multivariate analyses from line 121 and the word ‘MANOVA’ from line 191).
In Table 2. it would be informative to include the proportions of dominant and sub-dominant species (maybe in brackets or in separate columns), because percentages in this year/season pattern are not given in the text.
Some specific notes are listed below:
Line 100. in the title of Table 1. ‘trap-nights’ is written, but in the text the Authors use trap days. I suggest using only one.
Line 133. in the title of Table 2. I suggest using ‘Dominance pattern’.
In Figure 2., I suggest writing a title near the axis, e.g. ‘proportion (%)’. In addition, I suggest writing the letters (a-d) in figures, because the Authors refer to the figures in this manner (e.g. ‘Figure 2a’). e.g. the left diagram in Figure 2. could be marked with letter a.
After the above-advised corrections, I recommend the manuscript for publication.
Author Response
Rev#2 comments and answers
Thank you for the positive comments, and please find our answers below, point to point
Comment
The strength of the manuscript is that the reference problem of potential pest species is well-introduced, so the goals of the manuscript are clear. In addition, this work is country-wide and the control areas were chosen appropriately. However, in the introduction the agricultural land transformation is described (line 67-72), but the Authors did not give details (e.g. area) of the agricultural lands other than orchards and fruit plantations (e.g. cereal crops). The title of the paper refers to a wider range of habitats than those investigated in the study. In the discussion, many of the cited studies were conducted in crop-dominated areas and the conclusion in line 297-298 refers to the whole area of Lithuania. Therefore, I suggest including some information (e.g. area) on other agricultural fields (crops) in the introduction. Based on the title and the discussion, and in the light of the findings in this study, a short evaluation of similar studies in cereal fields of Lithuania would be relevant in the discussion. Alternatively, the Authors could give suggestions for future research in crop fields.
Answer
- Answering your comment, we added numbers for the sown area and area of meadows and pastures.
- We cannot add information on the small mammal research in the other agricultural habitats, as such investigations were not carried out in the country.
- Suggestion for such investigation added; thank you for this idea.
Text added in relevant parts of the manuscript.
Comment
The methods of data procession, the applied statistical methods are appropriate. However, Chi-square test is generally used to compare count data, and is not recommended to compare proportions. To test differences in percentage proportions, I recommend using G-test, or proptest in R. In the effect test of categorical predictor variables (e.g. crop age) in main effects ANOVA, I suggest to include the results of normality tests.
Answer
We re-calculated comparison of proportions using G-test (though, at the numbers on all calls > 5, chi-square may equally be used); obtained G-values did not change our conclusion). A
Normality of the dependent variable, relative abundance, was tested using Kolmogorov-Smirnov test. Obtained results were not uniform – we got some samples distributed normally, while other – not normally. Adding information on every test would be redundant, therefore we added some text to the chapter 2.3.
Afterwards, we analysed sensitivity of ANOVA to the normality of dependent variable. From the https://statistics.laerd.com/statistical-guides/one-way-anova-statistical-guide-3.php and other sources we accepted, that one-way ANOVA “ can tolerate data that is non-normal (skewed or kurtotic distributions) with only a small effect on the Type 1 error”. We checked residuals, also.
Then we re-calculated data using Kruskal-Wallis – this yielded similar result.
Therefore, finally we stay on the used method.
Comment
In the material and methods, the Authors wrote they used Hotelling’s T-square for multivariate analyses (MANOVA). However, in the section 3.2 (beginning in line 188), MANOVA results are not included (e.g. Hotelling’s T-square probabilities are not included). The results indicate that main effects ANOVA was used, and not multivariate ANOVA (MANOVA). This is correct, because if only one dependent variable was examined (proportions of M. arvalis), main effects ANOVA is appropriate. Therefore, I suggest changing the text accordingly (remove the description of Hotelling’s T-square for multivariate analyses from line 121 and the word ‘MANOVA’ from line 191).
Answer
Thank you for pointing this out; we corrected the text as advised.
Comment
In Table 2. it would be informative to include the proportions of dominant and sub-dominant species (maybe in brackets or in separate columns), because percentages in this year/season pattern are not given in the text.
Answer
We supplied these data, as recommended
Comment Line 100. in the title of Table 1. ‘trap-nights’ is written, but in the text the Authors use trap days. I suggest using only one.
Answer: corrected
Comment Line 133. in the title of Table 2. I suggest using ‘Dominance pattern’.
Answer: corrected as recommended
Comment: In Figure 2., I suggest writing a title near the axis, e.g. ‘proportion (%)’. In addition, I suggest writing the letters (a-d) in figures, because the Authors refer to the figures in this manner (e.g. ‘Figure 2a’). e.g. the left diagram in Figure 2. could be marked with letter a.
Answer: we supplied letters to the Figures 1–3. However, we find difficult to write Y axis titles, as in the Figure 3 it would be very long “Relative abundance (individuals per 100 trap days)”
Reviewer 3 Report
Review Comments: Diversity-1151859
Title: Common vole Microtus arvalis as a reference small mammal species in the Northern Zone
Authors: Vitalijus Stirkė , Linas Balčiauskas, Laima Balčiauskienė
Dear Authors,
I read your work with great interest and found your results very useful. Thank you for doing this work.
Overall, I have few substantive comments. The work seems largely well done and reported.
That said, I do have several issues with the way it is presented. Moreover, I offer suggestions on further analyses that could be done, however I do not see these as essential. Its just a suggestion.
Good luck revising your work.
SUBSTANTIVE COMMENTS:
1) Make the explict aims much clearer. What specifically is the goal of this work. While that is included in the Introduction, it could be made more clear with further elaboration. Some a priori predictions would help in this regard, especially given that the authors have published earlier on this topic.
2) Differentation and reference to similar work that was published earlier needs to be made very explicit. What does the current manuscript provide that is not already covered in the study in Animals, beyond two more years of data? I don’t see any problem here, however the previous work needs to be highlighted in the Introduction as well as how this work builds upon it. This work should also feature prominently and explicitly in the Discussion. (see: Balčiauskas, L.; Balčiauskienė, L.; Stirkė, V. 2019. Mow the grass at the mouse’s peril: diversity of small mammals in commercial fruit farms. Animals 9, 334.)
3) Ensure terminology is easily recognizable. Specifically, terms like “reference species”, “Northern zone”, and “territories” are not in common use in the literature, nor defined by the authors. For example, I had no idea what was meant by a “reference” species. It took some time to sort that out. Moreover, “focal” species is also used, apparently in the same context. I highly recommend using a more common term such as “indicator” species throughout the entire text, and ensuring that the reader knows what it is an indication of (plant protection practices).
4) Use common names throughout. The paper would be much more readable to audiences outside Europe and familiar with these species if common names were used throughout. I highly suggest replacing “M. arvalis” with “common vole” in the whole manuscript (except first usage). Same for the other species. I had to read very carefully so as to not mix up which species you were discussing, given M. arvalis and M. agrestis are so similar on paper. Make it easy for the reader, please.
5) I strongly urge you consider my suggestions regarding Tables 2 and 3 in the detailed comments.
6) I provide suggestions for reanalysis of the data based on a modeling (not ANOVA) approach that includes all covariates. I view this as an optional approach, but it would be a better use of the covariate data.
DETAILED AND EDITORIAL COMMENTS
Line 2: For me, the title is cryptic. I suggest something along the lines of:
“Evaluating the common vole (Microtus arvalis) as an indicator species in agricultural landscapes of the Northern Zone”
Line 10: What is a “reference species”. Please replace here and throughout with “indicator species”.
Line 11: See substantive comments above regarding use of common names.
Line 13: Period after “season”. Replace “pattern” with “The pattern”.
Line 32: Replace “some kind of” with “a”
Line 33: Is “agrolandscape” a word? Better to use “agricultural landscapes”?
Line 35: Replace “diversities” with “diversity”
Line 38: This sentence is better placed as the last sentence of the previous paragraph.
Line 39: Replace “specifically relating to the damaging of crops” with “including crop damage”
Line 39: Delete “vole” – redundant
Line 42: “pest species” of what? Incomplete sentence.
Line 46: Delete “non-arbitraily”
Line 47: Delete “even”
Line 57: See substantive comment above about use of “focal” and “reference” species.
Line 66: But But But. The fine paper in Animals by these same authors is very similar in objective to this manuscript. See my substantive comment above.
Line 67: Replace “Having in mind” with “Given”
Line 78: Why just common voles and not also field voles as the objective here?
Line 85: “unmoved” is a typo.
Line 89: Figure 1. A legend is needed. Otherwise it’s a nice figure.
Line 100: Table 1. Its not clear to me why these additional covariates are included. That should be more explicit in the text. Moreover, why does the control treatments vary? A stronger design would have been if they were the same. Finally, and critically, if including these covariates, then I believe that the abundance of common voles (and other species) should have more properly been modeled using something like a GLMM that also accounted for year and season as fixed effects. That said, I’m not sure these covariates are of interest for the present study and may only serve as a distraction, particularly given the earlier published work in Animals.
Line 102: Replace “investigation” with “trapping”
Line 111: Replace “processing” with “analysis”
Line 133: Table 2. I don’t think this is a particularly useful table. I would far rather see a table with the mean capture per 100 trap nights +/- SD for ALL species captured in each main treatment (crop, berry, control) for each of the three years. This would be much more useful and relevant than the present, qualitative table.
Line 142: Importantly, snap traps are poor at capturing shrews compared to pitfall traps. Please make sure this point is acknowledged in the Discussion.
Line 156: What is meant by “best represented”? More abundant, or found in a higher proportion of sites? Please be specific.
Line 160: This final sentence would fit much better after Line 146.
Line 167: Here and throughout. Figures 2 and 3 do not have “a-d” labels, so the text should not refer to “2a” or “3c”, for example. Global change required.
Line 194: Place all of these statistics in parentheses (brackets) please.
Line 215: What is meant by “quite scarcely registered”. Vague. Please specifically refer to “relative abundance” or “percentage of sites” or both.
Line 218: Here and elsewhere in this paragraph – what is meant by “territories”??? It is not the biological sense. I believe you mean sampling sites, but it is unclear. Please use another term and make a global edit to the paragraph.
Line 224: Replace “first species” with “common vole”
Line 231: Here and throughout the entire text, I suggest that rounding to the nearest percentage is more than sufficient for these data and would improve readability. For instance, replace “2.72%” with “3%”
Line 231: Underlining of M. arvalis is a typo.
Author Response
Rev#3 comments and answers
Comment
1) Make the explict aims much clearer. What specifically is the goal of this work. While that is included in the Introduction, it could be made more clear with further elaboration. Some a priori predictions would help in this regard, especially given that the authors have published earlier on this topic.
Answer
We extended text in the introduction, adding working hypothesis “Specifically, we tested whether common voles or field voles were widespread and abundant enough in fruit orchards, berry plantations and adjacent control habitats. Based on long term small mammal investigations in the country [1] and preliminary results of their trapping in the orchards [12], our working hypothesis stated that the distribution and abundance of field voles was not sufficient to classify this as a focal species.” Hope you will find this acceptable.
Comment
2) Differentation and reference to similar work that was published earlier needs to be made very explicit. What does the current manuscript provide that is not already covered in the study in Animals, beyond two more years of data? I don’t see any problem here, however the previous work needs to be highlighted in the Introduction as well as how this work builds upon it. This work should also feature prominently and explicitly in the Discussion. (see: Balčiauskas, L.; Balčiauskienė, L.; Stirkė, V. 2019. Mow the grass at the mouse’s peril: diversity of small mammals in commercial fruit farms. Animals 9, 334.)
Answer
In the former study we put emphasis on the small mammal presence in commercial orchards and diversity of their communities. At that time, Lithuania was not referred as the country of Northern Zone, and had no focal species of small herbivore animals for evaluation of the plant protection agents. Therefore, we had a different aim for this manuscript.
As shown in the Table 2, domination pattern in the investigated habitats was not stable, therefore, two additional years added new insights. We are sure, that there is a need to continue investigation of 1-2-3 more years to cover possible effects of the fluctuations of vole numbers.
Comment
3) Ensure terminology is easily recognizable. Specifically, terms like “reference species”, “Northern zone”, and “territories” are not in common use in the literature, nor defined by the authors. For example, I had no idea what was meant by a “reference” species. It took some time to sort that out. Moreover, “focal” species is also used, apparently in the same context. I highly recommend using a more common term such as “indicator” species throughout the entire text, and ensuring that the reader knows what it is an indication of (plant protection practices).
Answer
Change of the “reference species” to “focal species” was done throughout. We are thankful for this comment, as now changed term is compatible with EFSA documents. We, however, would not like change to “indicator species”, as voles do not provide information on the overall condition of the ecosystem and of other species in the orchard ecosystem. They hardly reflect the quality and changes in environmental conditions as well as aspects of community composition.
Comment
4) Use common names throughout. The paper would be much more readable to audiences outside Europe and familiar with these species if common names were used throughout. I highly suggest replacing “M. arvalis” with “common vole” in the whole manuscript (except first usage). Same for the other species. I had to read very carefully so as to not mix up which species you were discussing, given M. arvalis and M. agrestis are so similar on paper. Make it easy for the reader, please.
Answer
According the comment, we changed all scientific names to the common names in the text, except of the first usage.
Comment
5) I strongly urge you consider my suggestions regarding Tables 2 and 3 in the detailed comments.
Answer
This should be an misunderstanding, as we have only two tables in the text. If, however, Figures 2 and 3 were mentioned, changes were done according your previous comment.
Comment
6) I provide suggestions for reanalysis of the data based on a modeling (not ANOVA) approach that includes all covariates. I view this as an optional approach, but it would be a better use of the covariate data.
Answer
We analysed our data using GLMM with year and season as covariates. Results did not change our conclusions, and, as you wrote in the next comment related to Line 100, they are out of the scope of the manuscript.
Moreover, we would like to have bigger matrix for such analysis, adding 2 more ears and some new sites.
Comment Line 2: For me, the title is cryptic. I suggest something along the lines of: “Evaluating the common vole (Microtus arvalis) as an indicator species in agricultural landscapes of the Northern Zone”
Answer
According the comment above, we will use „focal species“. Also, commercial orchards did not stand for all agricultural landscapes.
Next, proposed Title is too long; we got a letter from Diversity, asking for a 8–13 word title.
Therefore, to acknowledge your comment, we propose new title as
Common vole as a focal small mammal species in orchards of the Northern Zone
Comment
Line 10: What is a “reference species”. Please replace here and throughout with “indicator species”.
Answer
Changed to “focal species”, to match with EFSA; see also answer above.
Comment
Line 11: See substantive comments above regarding use of common names.
Answer
All scientific names of the species to the common names throughout.
Comment Line 13: Period after “season”. Replace “pattern” with “The pattern”.
Answer: corrected
Comment Line 32: Replace “some kind of” with “a”
Answer: corrected
Comment Line 33: Is “agrolandscape” a word? Better to use “agricultural landscapes”?
Answer: corrected
Comment Line 35: Replace “diversities” with “diversity”
Answer: replaced
Comment Line 38: This sentence is better placed as the last sentence of the previous paragraph.
Answer: two paragraphs merged, thus, your comment acknowledged.
Comment Line 39: Replace “specifically relating to the damaging of crops” with “including crop damage”
Answer: replaced
Comment Line 39: Delete “vole” – redundant
Answer: deleted
Comment Line 42: “pest species” of what? Incomplete sentence.
Answer: “listed as main crop pest species”
Comment Line 46: Delete “non-arbitraily”
Answer: deleted
Comment Line 47: Delete “even”
Answer: deleted
Comment Line 57: See substantive comment above about use of “focal” and “reference” species
Answer: fully acknowledged; see answer above
Comment Line 66: But But But. The fine paper in Animals by these same authors is very similar in objective to this manuscript. See my substantive comment above.
Answer: In the former study we put emphasis on the small mammal presence in commercial orchards and diversity of their communities. At that time Lithuania was not referred as the country of Northern Zone, and had no focal species of small herbivore animals for evaluation of the plant protection agents. Therefore, we had a different aim for this manuscript.
Comment Line 67: Replace “Having in mind” with “Given”
Answer: replaced as suggested
Comment Line 78: Why just common voles and not also field voles as the objective here?
Answer: thank you, field vole added
Comment Line 85: “unmoved” is a typo.
Answer: “unmowed” – corrected
Comment Line 89: Figure 1. A legend is needed. Otherwise it’s a nice figure.
Answer: legend added, as well as letters a,b
Comment
Line 100: Table 1. Its not clear to me why these additional covariates are included. That should be more explicit in the text. Moreover, why does the control treatments vary? A stronger design would have been if they were the same. Finally, and critically, if including these covariates, then I believe that the abundance of common voles (and other species) should have more properly been modeled using something like a GLMM that also accounted for year and season as fixed effects. That said, I’m not sure these covariates are of interest for the present study and may only serve as a distraction, particularly given the earlier published work in Animals.
Answer: In the Table 1 we present trapping effort to show, that not all possible combinations of parameters were equally sampled. In 2020, we started to add under-sampled crops, ages or treatment intensities. Due to limited time of trapping and lack of sites with required characteristics it was (and still is) not possible to have full matrix of characteristics.
We did GLMM with season and year as covariates already, and this do not change our conclusions. However, as trapping is continued, we prefer to add 2021–2022 trappings to have better representation of not numerous species, and prepare publication with additional data. Results of GLMM, therefore, will not be included here. We may present some GLMM outcome to you on request.
Comment Line 102: Replace “investigation” with “trapping”; Line 111: Replace “processing” with “analysis”
Answer: replaced as suggested
Comment
Line 133: Table 2. I don’t think this is a particularly useful table. I would far rather see a table with the mean capture per 100 trap nights +/- SD for ALL species captured in each main treatment (crop, berry, control) for each of the three years. This would be much more useful and relevant than the present, qualitative table.
Answer: Table was intended to show pattern of dominance. We did not present all species trapping results, as the aim was to find which one of the two grey vole species is best suited to be a focal one. As for M. arvalis, abundance is presented in the Fig. 3 in all treatments.
Your comment, nevertheless, is acknowledged – we added proportion to each dominant and subdominant species in the Table 2.
Comment Line 142: Importantly, snap traps are poor at capturing shrews compared to pitfall traps. Please make sure this point is acknowledged in the Discussion.
Answer: text added to chapter 2.2 Small mammal trapping
Comment Line 156: What is meant by “best represented”? More abundant, or found in a higher proportion of sites? Please be specific.
Answer: changed to “We found common vole proportion being higher in the orchards and plantations than in the control habitats”
Comment Line 160: This final sentence would fit much better after Line 146.
Answer: changed according the comment
Comment Line 167: Here and throughout. Figures 2 and 3 do not have “a-d” labels, so the text should not refer to “2a” or “3c”, for example. Global change required.
Answer: done
Comment Line 194: Place all of these statistics in parentheses (brackets) please.
Answer: done
Comment Line 215: What is meant by “quite scarcely registered”. Vague. Please specifically refer to “relative abundance” or “percentage of sites” or both.
Answer: changed to “whereas field vole proportions were quite scarce.”
Comment Line 218: Here and elsewhere in this paragraph – what is meant by “territories”??? It is not the biological sense. I believe you mean sampling sites, but it is unclear. Please use another term and make a global edit to the paragraph.
Answer: changed to “sampling sites” throughout
Comment Line 224: Replace “first species” with “common vole”
Answer: replaced
Comment Line 231: Here and throughout the entire text, I suggest that rounding to the nearest percentage is more than sufficient for these data and would improve readability. For instance, replace “2.72%” with “3%”
Answer: not fully acceptable, as 0.35% (CI = 0.17–0.73%) would become 0% (0–1%) – this is not good. We, however, fully agree, that numbers may be presented with rounding to the first decimal, and followed your advice throughout the text.
Comment Line 231: Underlining of M. arvalis is a typo.
Answer: corrected